# Setting the boundaries–an approach to estimate the Loss Gap in dairy cattle

João Sucena Afonso[1,2]*, William Gilbert[1,2], Georgios Oikonomou[1], Jonathan Rushton[1,2]

1 Department of Livestock and One Health, Institute of Infection, Veterinary & Ecological Sciences, University of Liverpool, Liverpool, United Kingdom, 2 Global Burden of Animal Diseases Programme, University of Liverpool, Liverpool, United Kingdom

* joao.sucena-afonso@liverpool.ac.uk

**Data Availability Statement:** All relevant data are within the manuscript and its Supporting Information files. Also the data made available in the publication does not include any element that might compromise participant privacy.

## Abstract

Dairy production in the UK has undergone substantial restructuring over the last few decades. Farming intensification has led to a reduction in the total numbers of farms and animals, while the average herd size per holding has increased. These ever-changing circumstances have important implications for the health and welfare of dairy cows, as well as the overall business performance of farms. For decision-making in dairy farming, it is essential to understand the underlying causes of the inefficiencies and their relative impact. The investigation of yield gaps regarding dairy cattle has been focused on specific causes. However, in addition to the risk of overestimating the impact of a specific ailment, this approach does not allow understanding of the relative contribution to the total, nor does it allow understanding of how well-described that gap is in terms of underlying causes. Using the English and Welsh dairy sectors as an example, this work estimates the Loss Gap–composed of yield losses and health expenditure - using a benchmarking approach and scenario analysis. The Loss Gap was estimated by comparing the current performance of dairy herds as a baseline with that of scenarios where assumptions were made about the milk production of cows, production costs, market prices, mortality, and expenditure related to health events. A deterministic model was developed, consisting of an enterprise budget, in which the cow was the unit, with milking herd and young stock treated separately. When constraining milk production, the model estimated an annual Loss Gap of £148 to £227 million for the whole sector. The reduction in costs of veterinary services and medicines, alongside herd replacement costs, were important contributors to the estimate with some variation between the scenarios. Milk price had a substantial impact in the estimate, with revenue from milk yield representing more than 30% of the Loss Gap, when milk price was benchmarked against that of the top performing farms. This framework provides the boundaries for understanding the relative burden from specific causes in English and Welsh dairy cattle, ensuring that the sum of the estimated losses due to particular problem does not exceed the losses from all-causes, health or non-health related.

**Funding:** We would like to the thank the N8 Agri Food research partnership for having funded this work (University of Liverpool's award number JXG10710). For clarity and disclosure, it must be stated that the funders had no role in study design, data collection and analysis, decision to publish, or preparation of the manuscript.

**Competing interests:** The authors have declared that no competing interests exist

## Introduction

The intensification of livestock farming, facilitated by scientific and technological development, has reshaped the landscape of the dairy sector. Improved genetics, enhanced animal nutrition and access to technology have improved the efficiency of dairy systems in converting feed into outputs [1, 2]. On the other hand, the increase in efficiency has had an effect on the epidemiological profile of diseases affecting the productivity of dairy cattle, with intensive systems revealing a higher prevalence of production diseases such as mastitis and lameness [3, 4], which in turn can have an effect on animal health management strategy and its associated expenditure. The markets in which dairy farms operate also play a role. Price fluctuation can compromise the economic sustainability of dairy farms, thus reducing the resources for managing the dairy herd, particularly if in the main inputs, like concentrate feed, and outputs, like milk [5]. These ever-changing circumstances have important implications in the performance of the dairy units. The ability to understand the underlying causes of the inefficiencies of dairy systems in converting inputs into outputs, their relative impact, and the expenditure to ameliorate efficiency and mitigate these sources of inefficiency is essential for the decision-making process at farm-level, and other levels of the supply chain [6]. In the absence of such information, it is hard to unbiasedly prioritise existing problems and assess the economic efficiency of actions to mitigate their impact.

The investigation of yield gaps regarding dairy cattle have been focused on specific causes, be it nutritional management, genetic selection or health and welfare events [7, 8]. However, this approach can result in the overestimation of the impact of the specific ailment [9, 10]. Additionally, without a baseline providing the all-cause yield losses and expenditure to the dairy sector, it is hard to understand the contribution of each problem in the broader context of the gap. Therefore, it seems essential to define boundaries to this gap in a livestock sector, creating a framework for critical appraisal of the contribution of single-cause problems and expenditure. Furthermore, establishing such a "ceiling", ensures that the sum of the losses and expenditure due to different causes does not exceed the total losses and expenditure from all causes [11, 12]. The purpose of this exercise is not to remove the health hazards from the production system under study, but rather to evaluate the costs, in terms of production losses and health expenditure, due to the presence of such health hazards.

Using the English and Welsh dairy sectors as an example, we explore a benchmarking approach and scenario analysis to estimate the gap, hereafter referred to as the Loss Gap. Having had access to the distribution of dairy farms according to their economic net margin, and data on their production costs, revenues and key performance indicators, we compare the actual performance of these dairy systems with utopian scenarios, in which assumptions are made regarding milk production levels, production costs and prices through benchmarking. In this approach, the top-ranking farms serve as a reference and the lower-ranking farms are assumed to improve their performance in comparison to their higher-performing peers. The idea is that it is possible to improve the efficiency of a given business by setting goals according to the performance levels of the best organisations for that same business [13]. Additionally, further assumptions were made with regards to mortality and health expenditure.

This work proposes estimating the Loss Gap of the English and Welsh dairy sectors through benchmarking and removal of health expenditure and mortality, and appreciate the importance of different parameters, while discussing the applied methodology and alternatives.

## Materials and methods

### Ethics approval

Ethics approval ref: VREC769 was granted by the Veterinary Research Ethics Committee of the University of Liverpool.

### Analytical framework and data needs

The Loss Gap for the English and Welsh dairy sectors was estimated by comparing the current performance of dairy herds–baseline–with that of hypothetical scenarios. To establish a baseline, data are required for understanding the:

- ontological systems used for classifying animals and production units,

- animal population (size, age, yield),

- inputs and outputs of the different production units,

- distribution of study population across different production units,

- price system in which these units operate (Fig 1).

The efficiency of the sector in converting inputs into outputs is improved in the hypothetical scenarios by making assumptions about the milk production levels of the dairy population. This would capture the yield gap, by assessing the difference between the current performance of dairy system with that of scenarios with improved yield. Additionally, mortality and expenditure related with vet medication and services were set to zero. The intention is to create a disease-free context, where losses incurred from diseases and accidents and costs resulting from disease presence - expenditure with animal health–are removed. Lastly, and recognising the impact of the costs of key inputs and milk prices, milk prices and concentrate feed costs were assumed constant. In sum, depending on the scenario, the Loss Gap is composed of the yield gap, losses due to mortality, animal health expenditure, milk price and input cost difference. The considered scenarios are described in detail in the ***Model and software*** section.

### Data sources

Data sources with information on dairy farm production in the United Kingdom were identified and retrieved [14–20]. References were evaluated for their completeness and suitability for aggregation. In the Supplementary materials, S1 Table provides an overview of the data presented across the different sources, and S2 Table the descriptors for the different costs centres across the selected relevant references with data on farm enterprise budget. Due to the heterogeneity in methodologies and completeness across the different data sources, and different ways by which results were presented, it was not possible to make use of all the identified data sources. Whereas most of the data sources provided average figures for the different parameters across production systems, the Agriculture and Horticulture Development Board (AHDB) data offered more detailed information, as farms within each production system were grouped into quartiles according to their economic performance. Given the level of detail of data reported, and suitability in informing the model by how results were presented, a decision was made to use Agriculture and Horticulture Development Board (AHDB) as the main source of information. The AHDB data used related to the period 2018/2019. Thus, when possible, all figures used referred to that year, prices included. Where AHDB lack data, other references

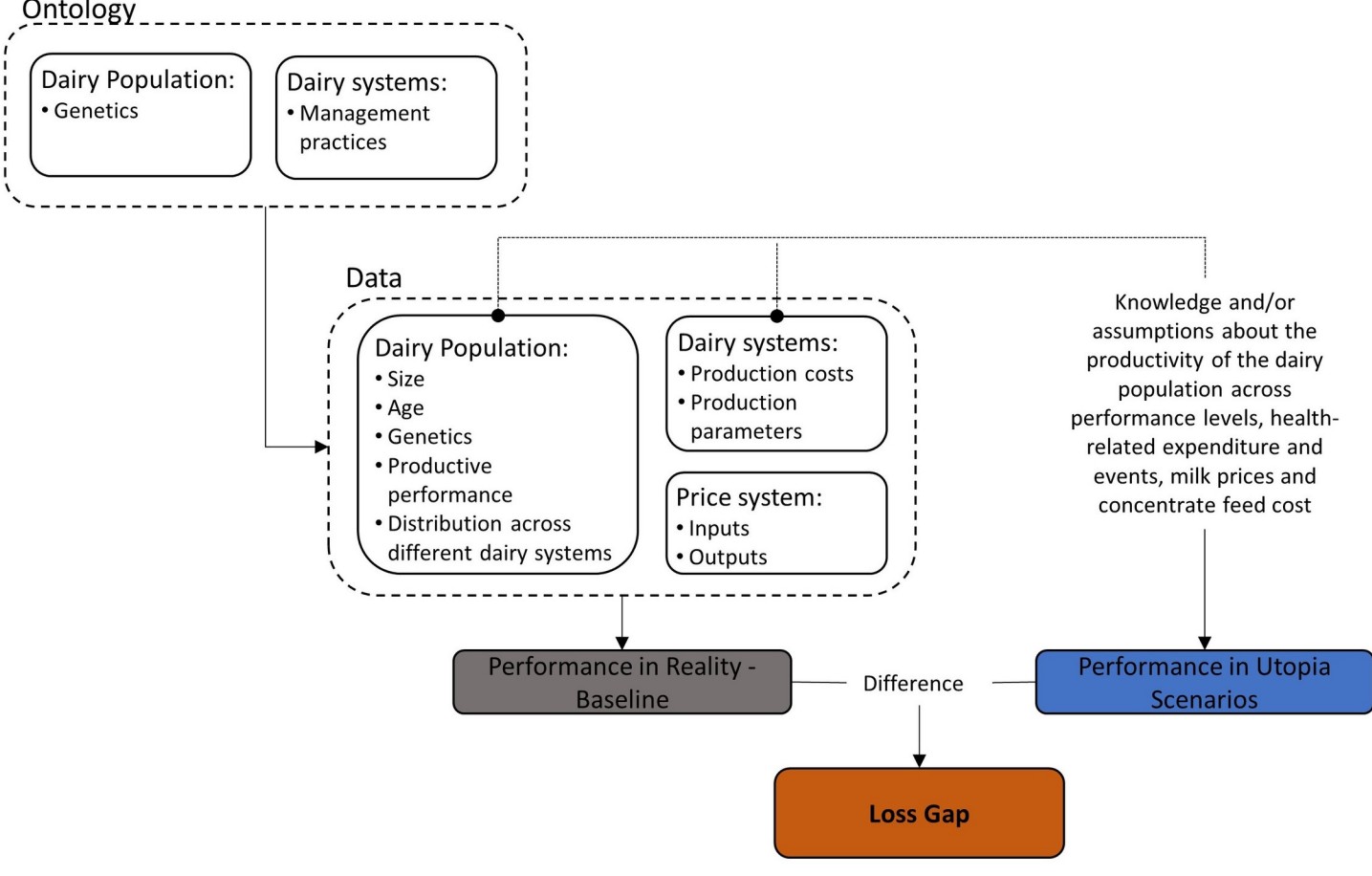

**Fig 1. Framework for estimating the Loss Gap of the English and Welsh dairy sectors.**

were used. Tables with the figures used and references for each parameter are presented in the *Model and software* section.

## Model and software

A cost analysis workbook was created in Microsoft Excel Professional Plus (2016) [21].

The milking herd and young stock were treated as distinct business activities [22], each with a spreadsheet. The model consisted of an enterprise budget without considering subsidies, in which the cow was the unit.

Taking into consideration the data availability and classifications used in the selected reference, this exercise focused on the three most representative dairy production systems according to the calving pattern–all year-round calving (AYRC), spring calving (SC) and autumn calving (AC) [14–16, 23].

Three performance levels within each dairy production system were also considered - top, middle and bottom–according to the categorisation of the farms in the AHDB report, for which quartiles on the full economic net margin (pence per litre) were used. The top performance level comprised the top 25% performing farms (from the third quartile upwards), the middle performance level included the farms in the interquartile range (between the first and the third quartiles), and the bottom performance level the bottom 25% performing farms [16], The document provided the averages for each parameter for the top and middle performance

categories across the different production systems. Data on the performance of the farms in the bottom category (poorer 25% performing farms) were retrieved following a request to the authors of the AHDB report on dairy performance. Farm enterprise budgets were estimated for each of the categories.

Three distinct counterfactual scenarios were created. The text below presents an overview of the assumptions for each.

- Utopia 1: milk yield per cow equal to that of the top performing levels across all production system; Zero mortality and no vet costs are assumed.

- Utopia 2: same as Utopia 1, with concentrate feed costs across the different performance levels equal to that of the top 25% performing level.

- Utopia 3: same as Utopia 1, with milk prices across the different performance levels equal to that of the top 25% performing level.

The enterprise budgets were used to estimate the performance of the English and Welsh dairy sectors according to the distribution of animals across the different production systems and performance levels. The Loss Gap was estimated as the difference between the enterprise budgets in the baseline and "Utopia" (Fig 1).

**Distribution of the dairy cattle population across the production systems, and performance level within each system for the baseline scenario.** Due to the lack of data on the number of animals farmed across all considered production systems and performance levels, the available data were used to estimate how many cows were kept in spring, autumn and AYRC calving systems, and in each performance level within each production system.

According to a survey conducted by AHDB, 81% of the farms in the UK are AYRC calving, 8% are AC and 4% are SC (the remaining 7% are dual block calving, as in having two calving windows, for example one spring another autumn) [23]. From the total number of English and Welsh dairy farms reported in 2019 [24], the number of farms in AYRC, AC and SC systems was estimated based on these proportions (Fig 2). The proportion of cattle in each performance category and production system was estimated from the AHDB data sample [16] and number of farms calculated as above, and then multiplied by the total number of dairy cows as reported by AHDB [25] to give the total number of animals in each (Fig 2).

**Milk production.** The model was developed keeping total milk production volume constant to the milk production officially reported by the UK for 2019. This was achieved by allowing for the cattle population to change across utopias while making the milk production equal to the one estimated for the baseline, using the *Goal Seek* function in Microsoft Excel Professional Plus (2016) [21].

**Milking herd parameters and calculations.** Table 1 presents an overview of the figures used in the model and the data sources they were retrieved from.

The following text provides an explanation on the methods by which estimates were calculated and assumptions made.

*Calf mortality*. The calf mortality parameter was informed by the figures reported in the John Nix pocketbook, the agricultural budgeting and costing book and by Hyde et al (2020) [14, 15, 18]. The calf mortality was estimated as the average of the figures provided in the three references. Additionally, it was assumed that the mortality rate was the same across all performance levels within each production system.

*Cow mortality*. The cow mortality parameter was informed by the figures reported in the John Nix pocketbook and the agricultural budgeting and costing book [14, 15]. The estimate was taken from the average of the figures provided in the considered references, and it was

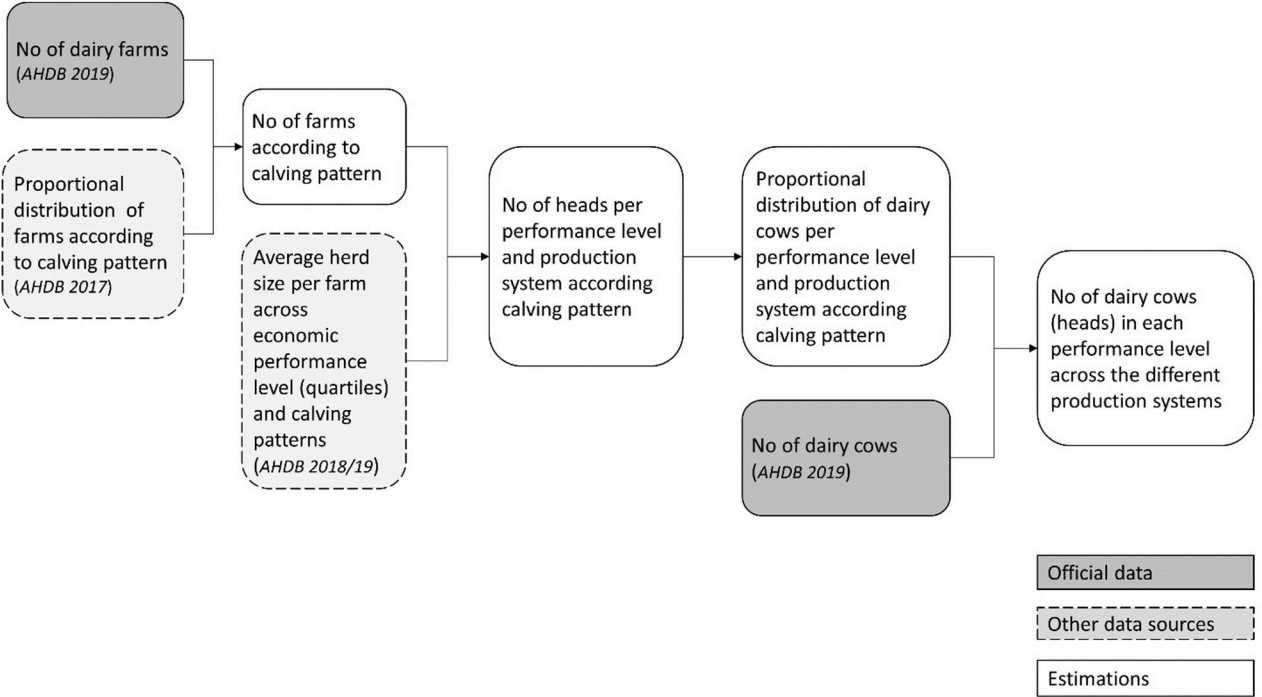

**Fig 2. Workflow for estimating the dairy population in each performance level across the different production systems.**

assumed that the mortality rate was the same across all performance levels within each production system.

**Young stock parameters and calculations.** Table 2 presents an overview of the figures used in the model and the data sources they were retrieved from. The different cost centres included the same parameters as the milking herd (e.g. *other feed and forage* cost centre includes fertiliser and lime, crop sprays, seeds and other crop costs).

The following text provides an explanation on the methods by which estimates were calculated, and underlying assumptions.

*Cow mortality*. The estimation and assumptions made for this parameter were the same as described for the milking herd (see *Milking herd parameters and calculations*).

*Heifer mortality*. The heifer mortality was obtained in the work by Boulton et al (2015). As with the cow and calf mortality, this rate was assumed to be the same across all performance levels within each production system.

*Number of replacements required*. This parameter was estimated by multiplying the *herd replacement rate* to the total number of animals within each performance category across all different production systems.

As the utopia scenarios assumed no mortality, the *cow mortality* was subtracted from the *herd replacement rate* to retrieve the number of animal replacement required.

**Prices.** The figures on the prices for the farms' outputs and their data sources are outlined in Table 3.

The methods for estimating each parameter, and assumptions made, are described in the text below.

*Milk price (pence per litre)*. This parameter was informed by data on dairy performance provided by AHDB [16].

**Table 1. Parameters for the milking herd model.**

| | Spring Calving | | | Autumn Calving | | | AYRC Calving | | |
|---|---|---|---|---|---|---|---|---|---|
| | Top 25% | Middle 50% | Bottom 25% | Top 25% | Middle 50% | Bottom 25% | Top 25% | Middle 50% | Bottom 25% |
| **Production parameters** | | | | | | | | | |
| Herd replacement rate (%)[3] | 27.5% | 26.1% | 28.4% | 24.4% | 28.3% | 25.9% | 27.7% | 29.8% | 25.1% |
| Calf mortality (%)[1, 2, 4] | 6.5% | 6.5% | 6.5% | 6.5% | 6.5% | 6.5% | 6.5% | 6.5% | 6.5% |
| Cow mortality (%)[1, 2] | 2.0% | 2.0% | 2.0% | 3.5% | 3.5% | 3.5% | 5.5% | 5.5% | 5.5% |
| Average milk yield (litres/cow/year)[3] | 5,656 | 5,392 | 4,728 | 7,550 | 7,808 | 8,119 | 8,749 | 8,396 | 7,161 |
| **Variable Costs/Operating Inputs** | | | | | | | | | |
| Concentrate and bulk feed (ppl*)[3] | 4.9 | 4.8 | 4.2 | 6.2 | 8.1 | 8.5 | 8.2 | 8.5 | 9.3 |
| Forage (ppl)[3] | 3.1 | 1.9 | 1.1 | 0.5 | 1.0 | 1.6 | 1.1 | 1.3 | 1.7 |
| Other feed and forage costs (ppl)[3]** | 1.5 | 2.0 | 1.7 | 1.1 | 1.5 | 1.2 | 1.1 | 1.3 | 1.2 |
| Vet services and medicines (ppl)[3] | 0.7 | 0.8 | 0.7 | 0.8 | 1.1 | 1.0 | 0.9 | 1.0 | 1.0 |
| Bedding (ppl)[3] | 0.5 | 0.9 | 0.7 | 0.4 | 0.6 | 0.9 | 0.8 | 0.7 | 0.8 |
| AI services and consumables (ppl)[3] | 0.5 | 0.6 | 0.4 | 0.4 | 0.5 | 0.3 | 0.4 | 0.5 | 0.5 |
| Recording, parlour consumables, sundries (ppl)[3] | 0.9 | 0.9 | 1.5 | 1.0 | 1.1 | 1.0 | 0.9 | 1.1 | 1.2 |
| **Fixed/Overhead Costs** | | | | | | | | | |
| Labour (including unpaid labour) (ppl)[3] | 4.2 | 4.8 | 6.9 | 4.0 | 4.4 | 5.8 | 3.8 | 4.7 | 5.3 |
| Power and machinery (ppl)[3]*** | 3.9 | 4.2 | 5.7 | 3.3 | 4.5 | 5.5 | 4.0 | 5.2 | 6.3 |
| General overhead costs (ppl)[3]**** | 4.7 | 5.9 | 7.9 | 4.0 | 4.4 | 4.5 | 3.8 | 4.2 | 5.5 |

[1]John Nix Pocketbook 50th edition [15]

[2]ABC Budgeting and Costing Book May 2020 [14]

[3]AHDB–dairy performance report 2018/2019 [16]

[4]Hyde et al, 2020 [18]

*pence per litre

**includes fertiliser and lime, crop sprays, seeds and other crop costs

***includes machinery and repairs, contracting, depreciation, electricity, fuel and insurances with vehicles

****includes property repairs, building depreciation, land rents, borrowing, water, telephone, insurances, and other operational costs

*Dairy female calf, dairy male calf, beef crossbreed female calf, beef crossbreed male calf and culled cow (£/head).* The different calf prices and culled cow value were estimated taking into account the figures reported by John Nix pocketbook and the agricultural budgeting and costing book [14, 15]. An average was taken from the figures provided, and it was assumed that the estimates were the same across all performance levels within each production system.

**Farm enterprise budget model.** The farm enterprise budget was estimated for each of the performance categories within each production system by deducting the costs of production (variable and fixed) and the herd replacement costs to the revenue from the farm outputs (milk, calves sold and culled cows). The enterprise budgets were then added to get the performance of the English and Welsh dairy sectors.

*Income from milk.* The milk revenue stream was dependant on average milk yield per cow, cow mortality and milk price. Cow mortality was assumed to occur uniformly across the lactation period. Dead cows therefore contributed with half lactation thus half of the considered average milk yield per cow per year.

$$Farm\ milk\ sales\ revenue\ (£) = Dairy\ population * Average\ milk\ yield\ per\ cow\ per\ year *$$
$$((1 - Cow\ mortality) + Cow\ mortality * 0.5) * Milk\ price$$

**Table 2. Parameters for the young stock.**

| | Spring Calving | | | Autumn Calving | | | AYRC Calving | | |
|---|---|---|---|---|---|---|---|---|---|
| | Top 25% | Middle 50% | Bottom 25% | Top 25% | Middle 50% | Bottom 25% | Top 25% | Middle 50% | Bottom 25% |
| Production parameters | | | | | | | | | |
| Heifer mortality (%)[4] | 8.8% | 8.8% | 8.8% | 8.8% | 8.8% | 8.8% | 8.8% | 8.8% | 8.8% |
| Cow mortality (%)[1,2] | 2.0% | 2.0% | 2.0% | 3.5% | 3.5% | 3.5% | 5.5% | 5.5% | 5.5% |
| Livestock purchases | | | | | | | | | |
| Cows and heifers (ppl)[3] | 0.11 | 0.29 | 0.94 | 0.25 | 0.61 | 1.21 | 0.52 | 0.95 | 1.64 |
| Variable Costs/Operating Inputs | | | | | | | | | |
| Concentrate and bulk feed (ppl)[3] | 0.88 | 0.78 | 0.85 | 1.02 | 1.08 | 0.79 | 1.18 | 1.59 | 1.51 |
| Forage (ppl)[3] | 0.03 | 0.03 | 0.01 | 0.01 | 0.05 | 0.01 | 0.06 | 0.11 | 0.08 |
| Other feed and forage costs (ppl)[3]* | 0.16 | 0.20 | 0.17 | 0.11 | 0.20 | 0.11 | 0.18 | 0.21 | 0.15 |
| Vet services and medicines (ppl)[3] | 0.10 | 0.15 | 0.13 | 0.13 | 0.15 | 0.10 | 0.15 | 0.16 | 0.13 |
| Bedding (ppl)[3] | 0.05 | 0.11 | 0.09 | 0.10 | 0.14 | 0.11 | 0.24 | 0.25 | 0.23 |
| AI services and consumables (ppl)[3] | - | 0.01 | 0.50 | - | 0.02 | 0.07 | 0.11 | 0.08 | 0.06 |
| Recording, parlour consumables, sundries (ppl)[3] | - | 0.01 | 0.52 | 0.25 | 0.07 | 0.15 | 0.29 | 0.40 | 0.21 |
| Fixed/Overhead Costs | | | | | | | | | |
| Labour (including unpaid labour) (ppl)[3] | 0.59 | 0.95 | 1.90 | 0.58 | 0.58 | 0.95 | 0.38 | 0.58 | 0.67 |
| Power and machinery (ppl)[3]** | 0.54 | 0.82 | 1.58 | 0.45 | 0.57 | 0.84 | 0.41 | 0.69 | 0.74 |
| General overhead costs (ppl)[3]*** | 0.67 | 1.16 | 2.21 | 0.60 | 0.57 | 0.72 | 0.39 | 0.53 | 0.66 |

[1]John Nix Pocketbook 50th edition [15]

[2]ABC Budgeting and Costing Book May 2020 [14]

[3]AHDB–dairy performance report 2018/2019 [16]

[4]Boulton 2015 [19]

*includes fertiliser and lime, crop sprays, seeds and other crop costs

**includes machinery and repairs, contracting, depreciation, electricity, fuel and insurances with vehicles

***includes property repairs, building depreciation, land rents, borrowing, water, telephone, insurances, and other operational costs

*Income from calf sales.* The revenue from calf sales depended on calf mortality, herd replacement rate and calf price. Additionally, it is assumed that the replacement animals required according to the replacement rate, and weighted for calf mortality, were retained in the farm.

**Table 3. Prices for farms' outputs.**

| | Spring Calving | | | Autumn Calving | | | AYRC Calving | | |
|---|---|---|---|---|---|---|---|---|---|
| | Top 25% | Middle 50% | Bottom 25% | Top 25% | Middle 50% | Bottom 25% | Top 25% | Middle 50% | Bottom 25% |
| Milk (ppl)[3] | 32.9 | 32.0 | 30.5 | 31.4 | 30.6 | 29.1 | 30.4 | 29.7 | 28.2 |
| Dairy female calf (£/head)[1,2] | 153 | 153 | 153 | 153 | 153 | 153 | 153 | 153 | 153 |
| Dairy male calf (£/head)[1,2] | 40 | 40 | 40 | 45 | 45 | 45 | 45 | 45 | 45 |
| Beef crossbreed female calf (£/head)[1,2] | 123 | 123 | 123 | 120 | 120 | 120 | 123 | 123 | 123 |
| Beef crossbreed male calf (£/head)[1,2] | 165 | 165 | 165 | 163 | 163 | 163 | 163 | 163 | 163 |
| Culled cow (£/head)[1,2] | 403 | 403 | 403 | 407 | 407 | 407 | 418 | 418 | 418 |

[1]John Nix Pocketbook 50th edition

[2]ABC Budgeting and Costing Book May 2020

[3]AHDB–dairy performance report 2018/2019

Furthermore, the birth of calves is evenly split between males and females.

$$Farm\ calf\ sales\ revenue\ (\pounds) = Dairy\ population*Herd\ replacement\ rate*$$
$$(1 + Calf\ mortality)*Price\ of\ male\ dairy\ calf + ((\ Dairy\ population - (2*$$
$$Dairy\ population*Herd\ replacement\ rate*(1 + Calf\ mortality))*(1-$$
$$Calf\ mortality)*Average\ price\ of\ beef\ calf)$$

*Income from culled cows.* The income obtained from culled cows was based on the value of the culled cow and the replacement rate, excluding the dead cows.

$$Farm\ culled\ cows\ revenue\ (\pounds) = Dairy\ population*(\ replacement\ rate - cow\ mortality)*$$
$$Culled\ cow\ price$$

## Results

### Milk production

The model estimated a total milk production of 10,087 million litres in the baseline scenario, an undersupply of 8.8% when compared to the reported milk production in 2019 by AHDB [26, 27].

### Dairy population

After subtracting the dairy cows reared in dual block calving systems, the baseline scenario considered a dairy cattle population of 1,285 thousand heads [25]. Out of the total population, 6.98% were farmed in spring calving systems. From the population within spring calving systems 29.78% were located in the 25% top performing farms, 57.12% in the middle performing farms and 13.10% in the 25% bottom performing farms. Out of the total population, 8.16% were farmed in AC systems. From the population within AC systems 28.44% were located in the 25% top performing farms, 49.07% in the middle performing farms and 22.50% in the 25% bottom performing farms. Out of the total population, 84.86% were farmed in AYRC calving systems. From the population within AYRC calving systems 34.23% were located in the 25% top performing farms, 48.27% in the middle performing farms and 17.50% in the 25% bottom performing farms. This meant that a third of the total dairy cattle population was reared in farms in the top performing category. The farms in the bottom performing category had the least number of dairy cows, accounting for 17.6% of the total population.

As milk production was assumed to be constant across all considered scenarios, taking as reference the milk production in the baseline scenario, the dairy population in the utopias was smaller to reflect the improvement in the production level as well as the absence of mortality. From the,285 thousand head in the baseline scenario, there was a reduction of 6.9% in the population. This represented a surplus of 89 thousand dairy cows for the considered utopias, in comparison to the total dairy population in the baseline scenario.

### Farm enterprise budget and Loss Gap

When considering the baseline scenario, milk sales represented the majority of the farm income at around 90% regardless of the production system or performance category. Apart from the bottom performing SC systems, feed (concentrate and forage) was the main cost for any dairy farm, ranging from 15.5% of the total costs in the bottom performing SC farms, to 35.5% in the high performing AYRC calving systems. Labour played an important role in the costs structure, representing between 14.5% in the top performing AYRC calving farms up to

20.5% of the total costs in the bottom performing SC farms. The general overhead costs and costs with power and machinery were also important inputs. Altogether, fixed costs (labour, power and machinery and general overhead) contributed with 44.1% of the total costs in the top performing AYRC calving farms up to 61.2% in the bottom performing SC farms. The costs of artificial insemination and consumables was the least significant across all farms, ranging from 1.0% in the bottom performing AC farms to 2.1% in the middle performing SC farms. In relation to the total costs, the young stock costs ranged between 3.7% in the top performing SC farms and 8.3% in the bottom performing SC farms (Table 4).

In the baseline scenario, with the exception of the SC farms, all bottom performing categories had a negative profit margin, estimated at £310; -£1,079 and -£55,924 thousand for the SC farms, AC farms and AYRC calving farms respectively. The category with the highest profit margin was the high performing AYRC calving systems at £185,985 thousand; followed by the middle performing AYRC calving systems at £59,779 thousand. The middle performing AC systems and top performing SC systems had roughly the same profit margin at around £16,000 thousand. The profit margin for the British and Welsh dairy sector in the baseline scenario was estimated at £264,447 thousand (S3 Table in the Supplementary materials).

The Loss Gap in the British and Welsh dairy sectors were estimated at £148,260, £186,203 and £227,086 thousand for Utopia 1, 2 and 3, respectively (Fig 3). This represents a loss per head of £115, £145 and £177 for Utopias 1, 2 and 3 respectively.

## Dissecting the Loss Gap

The percentages presented in Fig 4 reflect the variation of each cost centre from the baseline. This means that the variation for each cost centre from baseline to the Utopias can take a negative, a neutral (zero) or a positive value. Considering Utopia 1, the savings with veterinary services and medicines were the most significant contributor to the Loss Gap, with a variation of 65.8% of the estimate. Herd replacement costs and revenue from culled cows, resulting from

**Table 4. Percentage breakdown of revenues and of total costs per farm in the baseline scenario.**

| Parameters | Spring Calving | | | Autumn Calving | | | AYRC Calving | | |
|---|---|---|---|---|---|---|---|---|---|
| | Top 25% | Middle 50% | Bottom 25% | Top 25% | Middle 50% | Bottom 25% | Top 25% | Middle 50% | Bottom 25% |
| **Revenue** | | | | | | | | | |
| Milk | 91.4% | 90.9% | 89.1% | 93.5% | 93.3% | 93.4% | 94.2% | 93.7% | 92.7% |
| Calves | 3.5% | 3.9% | 4.2% | 3.1% | 2.7% | 2.9% | 2.5% | 2.4% | 3.5% |
| Culled cow | 5.1% | 5.2% | 6.7% | 3.4% | 4.0% | 3.7% | 3.4% | 3.9% | 3.9% |
| **Costs** | | | | | | | | | |
| Young stock | 3.7% | 4.6% | 8.3% | 4.2% | 4.4% | 4.6% | 4.6% | 6.1% | 4.9% |
| Concentrate and bulk feed | 19.0% | 17.1% | 12.4% | 27.2% | 28.4% | 26.8% | 31.1% | 28.1% | 26.9% |
| Forage | 11.9% | 6.7% | 3.2% | 2.1% | 3.6% | 5.2% | 4.3% | 4.3% | 4.9% |
| Vet services and medicines | 2.7% | 3.0% | 2.2% | 3.6% | 3.9% | 3.1% | 3.5% | 3.2% | 3.0% |
| Bedding | 1.9% | 3.3% | 2.0% | 2.0% | 2.1% | 2.9% | 3.1% | 2.3% | 2.2% |
| AI services and consumables | 1.8% | 2.1% | 1.3% | 1.7% | 1.8% | 1.0% | 1.5% | 1.6% | 1.4% |
| Recording, parlour consumables, sundries | 3.5% | 3.3% | 4.5% | 4.3% | 3.9% | 3.1% | 3.4% | 3.7% | 3.4% |
| Labour (including unpaid labour) | 16.2% | 17.0% | 20.5% | 17.7% | 15.4% | 18.2% | 14.5% | 15.4% | 15.3% |
| Power and machinery | 15.1% | 14.9% | 17.0% | 14.8% | 15.7% | 17.3% | 15.2% | 17.2% | 18.2% |
| General overhead | 18.3% | 21.0% | 23.6% | 17.5% | 15.6% | 14.1% | 14.4% | 13.9% | 16.0% |

Note: cells coloured in light grey are variable costs and cells coloured in light blue are fixed costs for dairy production

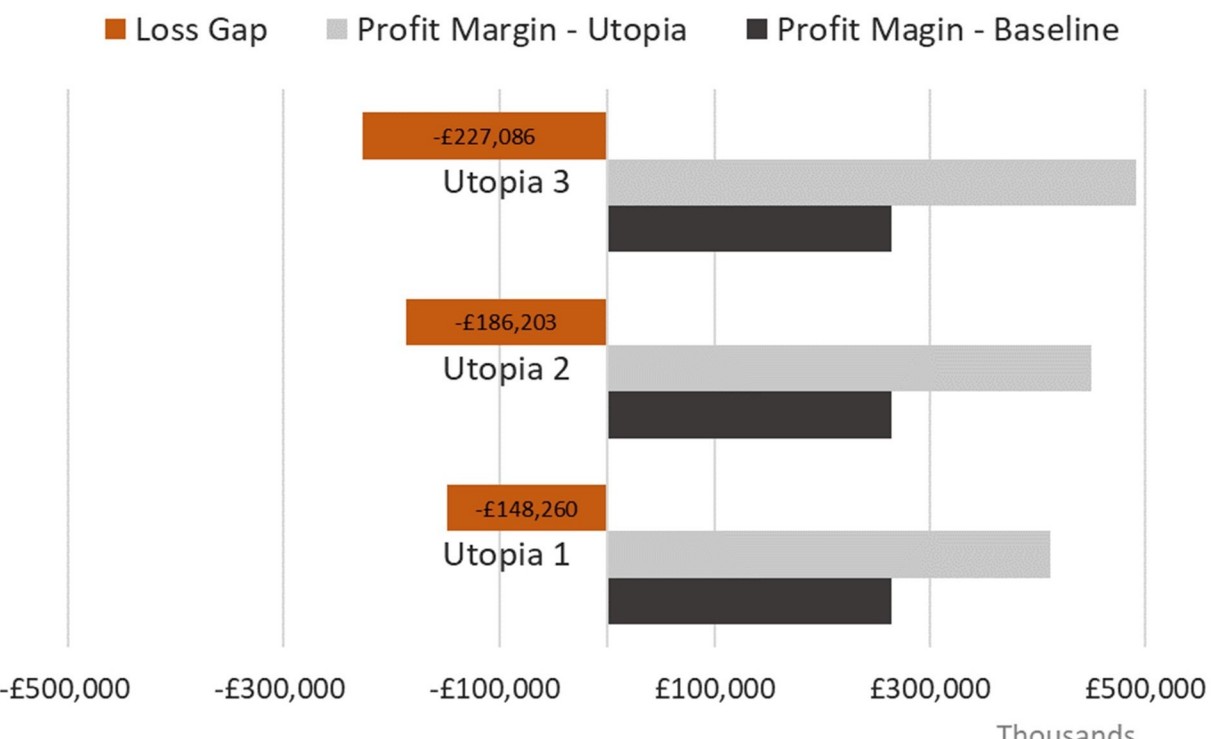

**Fig 3. Comparison between the enterprise budget of the British and Welsh dairy sectors in the baseline and utopias, and the associated Loss Gap for each.**

the absence of mortality, had the second and third most important variation - 34.3% and 11.4% respectively. The reduction in concentrate feed costs simulated in Utopia 2 led to a variation of 18.7% on the costs with these inputs, and to a reduction in the variation of the three parameters indicated above. In Utopia 2, savings with veterinary services and medicines, herd replacement costs and loss revenue from culled cows varied from the baseline by 52.4%, 27.3% and 9.1% respectively. This same phenomenon was observed in Utopia 3, where milk price was assumed to be that of the top performing level across all production systems. In Utopia 3, revenue from milk varied 33.1% from the baseline, whereas savings with veterinary services and medicines, herd replacement costs and revenue from culled cows varied 42.9%, 22.4% and 7.4% (Fig 4).

There were minor negative variations in several parameters when comparing baseline scenario with utopias, ranging from 3.3% for cost with power and machinery in Utopia 1 to less than 0.0% in bedding costs for all Utopias (Fig 4).

## Discussion

This paper presents an assessment of the yield losses and health-related expenditure, referred to as the Loss Gap, for the English and Welsh dairy sectors. The nature of the exercise is not prescriptive, but rather to test a methodology based on available data, and to foster discussion around a concept that the authors have recognised as much needed.

Different scenarios were considered for conducting the exercise. The Loss Gap for the English and Welsh dairy sectors was estimated at £148, £186 and £227 million for Utopias 1, 2 and 3, respectively. This meant that the inefficiencies from all-causes of these dairy systems in converting inputs into outputs, health or non-health related, ranged from about £148 million

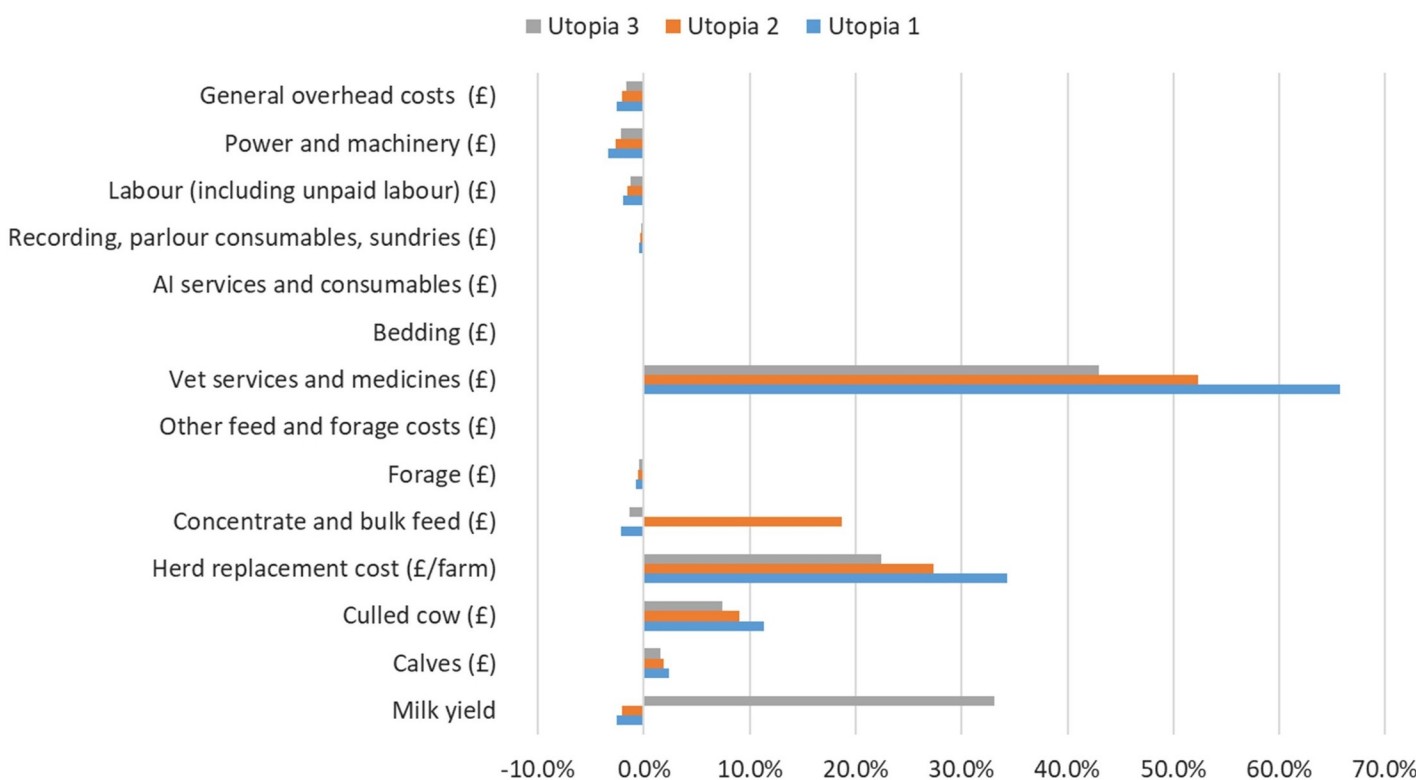

**Fig 4. Relative contribution of the different cost centres and revenue streams to the Loss Gap across different utopian scenarios.**

to £227 million. The savings with veterinary services and medicine costs, contributed for most of the Loss Gap regardless of the utopia. This was due to assuming a scenario where there was no need for animal health care. Costs avoided from animal replacements, were also major contributors to the Loss Gap regardless utopia. This was the result of reducing the number of animal replacements needed by bringing mortality to zero, and allowing the dairy herd to reduce in size as a result of improved yield per cow when constraining the model for total yearly milk produced. The observed variations in the contribution of these costs across the different utopias is justified by the different assumptions made for each utopia. In Utopia 2 the costs with concentrate feed per litre were made equal as those reported for the top performing farms, across all levels. Apart from the SC systems, these costs were increasingly lower from the bottom to the top performing levels. As the majority of the animals were farmed in AYRC systems, this led to an increase of the contribution of these costs to the Loss Gap, which brought the contribution of other costs or revenue streams down. A similar pattern was observed in Utopia 3 with an increase in the contribution of milk yield to the Loss Gap, as a result of milk price being made standard across all levels having as reference the top performing farms.

The contribution of revenue from culled cows reflects the smaller herd size resulting from increased milk yield per cow and also the reduction of mortality to zero. With fewer animals culled this revenue is smaller in utopias having a negative variation compared to baseline from 7.4% in Utopia 3 to 11.4% in Utopia 1. This was also the case for sales with calves, with the reduction in herd size leading the fewer animals being sold.

As observed in Fig 4, there were minor negative variations in several parameters when comparing baseline scenario with utopias, suggesting that the costs associated with the parameters were lower in the baseline when compared to the utopias. These were mainly due to how milk

production was distributed across the different performance levels. The bottom performing levels had higher milk production in the utopias compared to the baseline. This was the opposite for the middle and top performing levels, where production costs per litre were lower. Thus, this marginal increase in milk production in levels where production costs per litre were higher led to the observed negative variation between baseline and utopias (Fig 4).

Bennett and IJpelaar (2005) estimated the economic impact of a selection of cattle endemic diseases in the UK. Considering fifteen health conditions (mastitis, parasitic bronchitis, paratuberculosis, pasteurellosis, salmonellosis, summer mastitis, tuberculosis, bovine viral diarrhoea (BVD), *Escherichia coli*, enteric disease, fascioliasis, infectious bovine keratoconjunctivitis (IBK), infectious bovine rhinotracheitis (IBR), lameness, leptospirosis) they calculated yield losses and expenditure between £256 to £598 million every year [7]–a range of values beyond the highest estimate (£227 million) of the Loss Gap. According to Bennett and IJpelaar (2005), and in relation to the Loss Gap, the economic impact of these disease in cattle in the UK represents between 113%, considering the lowest estimate by the authors and Utopia 3, to 403% considering the highest estimate by the authors and Utopia 1. It is true that the authors estimate covers the whole of the UK, whereas the Loss Gap has been estimated for the English and Welsh dairy sectors, which could increase our estimate of the gap. Additionally, their estimates are for cattle diseases and not specifically for dairy cattle, and are based on outdated data, in particular that of prices. Nevertheless, this framework places a question mark on the extent to which co-morbidities are playing a role when estimating the impact of each disease, which could be driving the total impact beyond the limits of the Loss Gap.

The estimation of the Loss Gap is drawn by comparing the current reality of the sector–baseline - with that of utopian scenarios, in which health events were assumed to be non-existent and yield to expectedly improved. One of the challenges is how to define a healthy or a reference farm/animal. Understanding how utopias were built is vital as it can have a significant impact on the Loss Gap. When deciding on the method for defining Utopias, three approaches were considered and explored:

- the best farm within each system, as described in the empirical data;

- the best performing combination of parameters across all farms within each system;

- and a zero-disease circumstance (no disease and zero mortality) and maximised animal performance only limited by biology.

Using benchmarking to define the reference production unit seems a reasonable approach as this tool is used across the sector for comparison [20, 28]. It also delivers temporal context to the estimate by providing the "best performance under current circumstances". Although tempting, the second approach brings a problem hard to solve–the dependency of some production indicators. For example, high yielding cows have fertility problems in what seems to be a biological restriction of the genetic selection programmes' success [29]. Therefore, when cherry picking the best figure for each production parameter individually, this interconnection would be ignored. For this reason, a decision was made establish the reference production unit based on the existing empirical data. The third approach provided the grounds for defining Utopia 1 –a dairy sector in which all animals perform as the top reference unit according empirical data, which still considers the effect of disease constraining animal performance, in a context without disease (no veterinary nor medicine costs) nor mortality. Thus, disease effects are artificially removed by bringing mortality and expenditure with vet services and medicine costs to zero. As an exploratory exercise, it must be said that none of these choices are incorrect and all can provide useful information. Additionally, the developed utopias are limited while attempting to simulate a production system without diseases. As explained above, the milk

yield is based on empirical data, and thus the "gold standard" is still hampered by disease impact. Thus, it could be that the Loss Gap is underestimated. For example, the study by Hanks and Kossaibati (2020) indicate that cows from the best 25% farms considering yearly milk yield per cow of the 500 farms in their sample, which are also affected by health issues, produce at least 9,850 kg of milk per year [20]–a 12.5% difference from the average 8,749 kg milk/cow/year used in this study. Having identified the Loss Gap as a critical framework for the process of estimating the economic impact of health events in livestock production, the discussion around this topic is vital for its success as a replicable tool. If the model had allowed for the animal population to be stable, the milk production would increase. Modelling the ripple effects of a production surplus can be complex. Estimating the Loss Gap by constraining milk production avoided conducting such exercise. Additionally, estimating the dairy herd necessary for a stable milk production in a utopian scenario provides an indicator that can easily be translated into land use and environmental impact, which could be helpful when answering questions such as: what is the surplus of land currently used to meet the demand of milk and dairy products? What could be resulting environmental benefits of increasing the efficiency the dairy herd, assuming a constant milk production? Considering all of the above, a decision was made to design the model for estimating the Loss Gap by constraining milk production, allowing the dairy cattle population to vary from the baseline.

When allowing for a reduction of the dairy herd population to maintain constant milk output, and accepting a reduction in stock numbers at farm level, it seems unavoidable that some farmers would go out of business. Taking the average number of animals per dairy herd in the UK in 2020 [30], the improvement in the efficiency of the dairy systems would led to a reduction of 574 dairy farms. The farmers and employees were assumed to be absorbed by the job market. In fact, the number of farms in England and Wales have reduced over the last decades, from 28,093 in 1995 to 8,735 in 2019 [24]. According to the AHDB report, a substantial proportion of dairy farms is economically unviable [16], which means that government subsidies are essential for these families and their businesses to survive. This is particularly important given the discontinuation of the direct payment schemes as of 2021 as they were deemed to be a poor financial support tool and a distorted incentive to production [31].

Different data sources on the productive performance of dairy cattle and associated production costs are available. Some of the most comprehensive are based on agriculture surveys and have been published for decades [14, 15]. However, most of the explored references present data as averages without any information on the distribution measures. Knowing how the population is distributed across the different performance indicators is important for the creation of the utopian scenarios, as animals or farms with lower productive performance are assumed to improve through benchmarking. However, the AHDB report on dairy performance presents the data as averages per quartiles [16], and it was possible to retrieve data on the bottom quartile. Additionally, according to the publication, the sampling strategy was designed so that sample would be representative of the whole British dairy sector. For the reason outlined above, and given that the data available in this reference offered a better fit for how utopias were to be defined, a decision was made to make use of this reference as the main data source for this exercise. Limiting the data sources to only one reference might have increased the risk of introducing bias to the results, especially if the sampling strategy failed to produce a representative sample from the population. Additionally, analysing a subset of whole population and using averages from British dairy cattle herd might introduce errors in the estimations, as it could be that the subset of animals that are analysed are systematically different from the average animal. It is important to be aware of such data issues in order to challenge the validity of the results.

The reference used to inform the model classified farms according to their profitability, and not on the productivity of the animals. This challenged the estimation of the Loss Gap. It could be that farmers in the top performing level were being able to spread their fixed costs across a larger herd (effect of economies of scale), or it could be that they were able to purchase feed at a lower price compared to their peers. Additionally, by increasing milk yield per cow based on benchmarking, the comparisons drawn ignore certain factor that can influence dairy cattle yield, such as genetics, feed quality, stockmanship, infrastructure, investment in disease prevention and control. It would have been good to have had the opportunity to explore the correlation/interdependency of certain factors such as stocking density per land area and forage costs, as these could shed some light on the limitation of the production units in improving their performance. For example, it could be that some farms are performing better compared to their peers as a result of higher stocking density. It could also be that the land quality of certain farms limits stocking density. It would have been good to understand the volumes of inputs. This information, alongside the information on total costs with each input would allow to estimate the average costs per unit of volume and understand the influence of prices in the performance of the production units. Yet, and stated before, the lack of data hampers any adequate analysis.

Standardising for milk price has led to a significant increase in the Loss Gap estimate (from £148,260 in Utopia 1 to £227,086 in Utopia 3, a 53.2% increment) (S4 Table in the Supplementary materials). If on the one hand the difference in the milk price across different farms relates to quality (lower somatic cell count), on the other it can capture access to better milk contracts not necessarily related with milk quality. Price variations across the study population can affect significantly the Loss Gap estimate and this should be taken into consideration when interpreting the results.

Three utopian scenarios were defined as counterfactual to estimate the Loss Gap, all considering the average milk yield across all farms to be equal to the that observed in the top performing farms. Utopias 2 and 3 were developed to assess the impact of a key production factor–concentrated feed in Utopia 2 –and a key market parameter–milk price in Utopia 3 - in the Loss Gap. This has shed light about the impact of efficient use of key inputs and prices in the Loss Gap. Yet, the approach taken requires caution when interpreting any of the results across the different scenarios. The way utopias have been established, by improving cows' milk yield, could potentially mean the reallocation of animals to different geography (better land and climate) and assume different genetics. In the absence of data, it is difficult to ascertain which part of health-related issues explains the variation in the productive capacity. The absence of data is a key findings of this exercise, a problem that has been highlighted in literature [7, 32]. The easily accessible data lacks the necessary detail to conduct robust economic impact studies, and the information held by the private sector that could inform such assessment with precision is rarely available. As technology offers part of the solution for the data quality and quantity [33–35], other steps must be taken to reduce the gap. A recommendation would surely be to foster collaborative research approaches between scientists, public and private sectors, facilitating data sharing while making sure that data protection regulation is respected. For the time being, data will never be perfect in reliability and availability, therefore there will always be uncertainty around the estimates. As data becomes available, in quantity and quality, models can be better informed helping to reduce uncertainty and increase the precision of the produced results.

It must be acknowledged that the predicates of the dairy sector vary with time, be it in terms of the market, animal production and/or farm management. Consequently, the Loss Gap is a mutating estimate per definition and needs to be revisited and updated to reflect the changes to the sector.

This analysis is the result of an adjustment to the original approach described by Afonso, S. J. (2021) [36], which resulted in a substantial reduction in the Loss Gap estimate. In the original work, the improvements between performance levels were considered beyond solely milk yield, making assumption about inputs such as feed and labour costs, and other issues like stocking density which could related to farm management and/or soil characteristics. It was considered that this could mean changing the nature and/or location of the production systems as they currently exist. Additionally, we could be capturing figures that are unrelated to health and welfare, such as access to better feed prices. Thus, and while attempting to isolate yield losses and expenditure in animal health, the initial approach was adjusted to the one outlined in this paper.

This assessment is part of research work on the economic impact of hoof health in the English and Welsh dairy sectors. For that reason, a decision was made to work out the Loss Gap for these two dairy cattle populations. However, and regardless of the scope of the assessment, the purpose of the study was to propose, test and discuss a methodology for estimating the Loss Gap, which can be systematically replicated.

## Conclusion

When constraining the milk production, the model estimated the Loss Gap to range from about £148 million to £227 million. This framework provides the boundaries for assessing the relative economic impact from specific causes in the study population under investigation, ensuring that the sum of the estimated losses due to particular health disorders do not exceed the losses from all-causes, health or non-health related. The proposed approach is exploratory in nature, and should encourage further constructive discussions to improve methods, and identify potential solution for recognised data challenges.

## Supporting information

**S1 Table. Classification system and cost centres descriptors across the selected relevant references.**
(DOCX)

**S2 Table. Descriptors for the different costs centres across the selected relevant references with data on industry enterprise budget.**
(DOCX)

**S3 Table. Farm enterprise budget for each category of the production systems (results in thousands).**
(DOCX)

**S4 Table. Loss Gap estimate for different models.**
(DOCX)

## Acknowledgments

This article was based on work published as part of a PhD thesis [36] with important modifications to the original approach. The thesis is available in the repository of the University of Liverpool. The authors would like to thank AHDB for their assistance in making the data for conducting this assessment available.

## Author Contributions

**Conceptualization:** João Sucena Afonso, William Gilbert, Jonathan Rushton.

**Formal analysis:** João Sucena Afonso.

**Funding acquisition:** Jonathan Rushton.

**Investigation:** João Sucena Afonso.

**Methodology:** João Sucena Afonso.

**Project administration:** João Sucena Afonso.

**Supervision:** William Gilbert, Jonathan Rushton.

**Writing – original draft:** João Sucena Afonso.

**Writing – review & editing:** João Sucena Afonso, William Gilbert, Georgios Oikonomou, Jonathan Rushton.

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
