## [Decision Letter · Decision Letter 0]

24 Apr 2024

PONE-D-24-02553Setting the boundaries – an approach to estimate the yield gap and health expenditure in dairy cattlePLOS ONE

Dear Dr. Sucena Afonso,

Thank you for submitting your manuscript to PLOS ONE. After careful consideration, we feel that it has merit but does not fully meet PLOS ONE’s publication criteria as it currently stands. Therefore, we invite you to submit a revised version of the manuscript that addresses the points raised during the review process.

**This paper presents a deterministic model (method) of  estimating the Loss Gap of the English and Welsh dairy sectors through benchmarking. The paper is well written. However, authors have to attend to the minor comments that have been raised by reviewer number 1, 2 and 3.**

We look forward to receiving your revised manuscript.

Kind regards,

Chisoni Mumba

Academic Editor

PLOS ONE

Journal Requirements:1. When submitting your revision, we need you to address these additional requirements. Please ensure that your manuscript meets PLOS ONE's style requirements, including those for file naming. The PLOS ONE style templates can be found at https://journals.plos.org/plosone/s/file?id=wjVg/PLOSOne_formatting_sample_main_body.pdf and https://journals.plos.org/plosone/s/file?id=ba62/PLOSOne_formatting_sample_title_authors_affiliations.pdf 2. We note that the grant information you provided in the ‘Funding Information’ and ‘Financial Disclosure’ sections do not match.  When you resubmit, please ensure that you provide the correct grant numbers for the awards you received for your study in the ‘Funding Information’ section. 3. Thank you for stating the following financial disclosure: The work was funded by the N8Agrifood Please state what role the funders took in the study.  If the funders had no role, please state: "The funders had no role in study design, data collection and analysis, decision to publish, or preparation of the manuscript." If this statement is not correct you must amend it as needed. Please include this amended Role of Funder statement in your cover letter; we will change the online submission form on your behalf. 4. We noted in your submission details that a portion of your manuscript may have been presented or published elsewhere. This article was based on work published as part of a PhD thesis with important modifications to the original approach. Please clarify whether this [conference proceeding or publication] was peer-reviewed and formally published. If this work was previously peer-reviewed and published, in the cover letter please provide the reason that this work does not constitute dual publication and should be included in the current manuscript. 5. We note that your Data Availability Statement is currently as follows: All relevant data are within the manuscript and its Supporting Information files Please confirm at this time whether or not your submission contains all raw data required to replicate the results of your study. Authors must share the “minimal data set” for their submission. PLOS defines the minimal data set to consist of the data required to replicate all study findings reported in the article, as well as related metadata and methods (https://journals.plos.org/plosone/s/data-availability#loc-minimal-data-set-definition). For example, authors should submit the following data: - The values behind the means, standard deviations and other measures reported;- The values used to build graphs;- The points extracted from images for analysis. Authors do not need to submit their entire data set if only a portion of the data was used in the reported study. If your submission does not contain these data, please either upload them as Supporting Information files or deposit them to a stable, public repository and provide us with the relevant URLs, DOIs, or accession numbers. For a list of recommended repositories, please see https://journals.plos.org/plosone/s/recommended-repositories. If there are ethical or legal restrictions on sharing a de-identified data set, please explain them in detail (e.g., data contain potentially sensitive information, data are owned by a third-party organization, etc.) and who has imposed them (e.g., an ethics committee). Please also provide contact information for a data access committee, ethics committee, or other institutional body to which data requests may be sent. If data are owned by a third party, please indicate how others may request data access. 6. When completing the data availability statement of the submission form, you indicated that you will make your data available on acceptance. We strongly recommend all authors decide on a data sharing plan before acceptance, as the process can be lengthy and hold up publication timelines. Please note that, though access restrictions are acceptable now, your entire data will need to be made freely accessible if your manuscript is accepted for publication. This policy applies to all data except where public deposition would breach compliance with the protocol approved by your research ethics board. If you are unable to adhere to our open data policy, please kindly revise your statement to explain your reasoning and we will seek the editor's input on an exemption. Please be assured that, once you have provided your new statement, the assessment of your exemption will not hold up the peer review process. 7. Your ethics statement should only appear in the Methods section of your manuscript. If your ethics statement is written in any section besides the Methods, please move it to the Methods section and delete it from any other section. Please ensure that your ethics statement is included in your manuscript, as the ethics statement entered into the online submission form will not be published alongside your manuscript.  8. Please include captions for your Supporting Information files at the end of your manuscript, and update any in-text citations to match accordingly. Please see our Supporting Information guidelines for more information: http://journals.plos.org/plosone/s/supporting-information.  9. 
Please review your reference list to ensure that it is complete and correct. If you have cited papers that have been retracted, please include the rationale for doing so in the manuscript text, or remove these references and replace them with relevant current references. Any changes to the reference list should be mentioned in the rebuttal letter that accompanies your revised manuscript. If you need to cite a retracted article, indicate the article’s retracted status in the References list and also include a citation and full reference for the retraction notice.

Reviewers' comments:

Reviewer's Responses to Questions

**Comments to the Author**

1. Is the manuscript technically sound, and do the data support the conclusions?

Reviewer #1: No

Reviewer #2: Yes

Reviewer #3: Yes

2. Has the statistical analysis been performed appropriately and rigorously? 

Reviewer #1: No

Reviewer #2: Yes

Reviewer #3: Yes

3. Have the authors made all data underlying the findings in their manuscript fully available?

Reviewer #1: Yes

Reviewer #2: Yes

Reviewer #3: Yes

4. Is the manuscript presented in an intelligible fashion and written in standard English?

Reviewer #1: Yes

Reviewer #2: Yes

Reviewer #3: Yes

5. Review Comments to the Author

**Reviewer #1: **This article is basically about descriptive statistics of dairy farms, and I think this is not enough for a scientific publication. Moreover, the authors made scenarios which are totally unrealistic. Assuming no mortality and no vet costs is really unrealistic for dairy herds. At dairy herds, production diseases (eg mastitis, lameness, metabolic disorders) will always retain as they can't be eradicated. Comparing results to these utopian scenarios (no mortality and no vet costs) is thus extremely overestimating the losses. As the model is deterministic it does not allow to give insights in variation, which is a weak point. Only average values are taken into account.

small remark: replacement herd is a strange term. I think the authors mean young stock herd.

**Reviewer #2**: This study mainly aimed to estimate the loss gap regarding milk yield losses and health/veterinarian expenditure, as a holistic (Global) approach, in dairy cattle industry in United Kingdom. For this purpose, a basal model was established as basal scenarios, using data recorded from different sources which estimated the values of the input variables. Three types of farms were considered according to the annual calving pattern and were classified as a percentile 25% (top), >25 <75% and 75% or more. In a second step, 3 different utopic scenarios were made and compared with the baseline scenarios. This approach would to give a global economic assessment to estimate what is the more profitable model (lower loss gap) to prevent inefficiencies regarding the different assumptions and parameters used in this work.

Overall, this manuscript was well done and no major issues were found. The base line scenario seems to represent an acceptable estimation with a delay of about 4-6 years (data from 2018/2020). The counterfactual scenarios were made according the top 25% and considering zero mortality/health (vet) costs. These parameters are objective. But a more detailed justification and limitation for these 3 counterfactual scenarios can elucidate the readers. Also, a critical discussion about the limitations and constraints of this study is welcome. As example, the potential effect of farm size in each production system regarding incomes (production cost) and outcomes; or the cost of disease prevention in farms. Just a curiosity: if we consider the top 10% as reference what is the annual Loss Gap?

**Reviewer #3**: I agree with the authors that this work begins to provide a useful framework for assessing economic impacts and enjoyed reviewing this manuscript.

I have some comments on the manuscript most of which are minor but a couple of points I feel are worthy of revision.

Major comments:

Line 354 the section describing 'Dissecting the loss gap' needs some clarification. What do the given percentages represent, variation to baseline scenario? For utopia scenario 1, addition of 65.8%, 34.2% and 11.4% is 111.4%?

In relation to this point I think it should be made clearer in the manuscript that for example a variation of 65% from base on veterinary services and medicines when they are at 1 ppl only gives a modest change in cost expressed as ppl whereas a 35% variation in herd replacement costs would be a much bigger change in ppl.

Line 447-450 In utopia 1 the yield level achieved by the best performing herds is still constrained by disease but most likely at a much lower level. Top level performance here is not 'disease free'. I think this needs to be stated. The Loss Gap would be much greater if in the scenarios modelled a top performing yield level that was genuinely disease free. It is a limitation of this study that you cannot make this true disease free comparison. For example the NMR 500 (https://www.nmr.co.uk/news/nmr-500-herd-kpi-2023-report-released) herds reports a much higher yield level and could be cited as evidence. This would also suggest that the loss gap estimated in this current study is also under estimated and gives an indication of how big the under estimate is. The NMR 500 report however does not provide all the figures necessary to estimate a loss gap using the methods reported.

Minor comments:

Lines 1-4 and elsewhere e.g. line 70 uses the phrase 'yield gap' whilst the Abstract and rest of the manuscript uses 'Loss Gap'. This is slightly misleading as I think what is being evaluated here is the difference in gross margin and would suggest to use the term 'Loss Gap' throughout.

Line 58 or is the higher prevalence due to a survivor bias i.e. lower yielding cows disposed of earlier in a disease episode? And does this mean that earlier culling of those cows affects the Loss Gap. I'm guessing no information is available on cow yield at exit but interesting to speculate.

Line 75 'of the gap'

Line 148 uses the abbreviation AYRC while at line 180 it is AYR.

Line 294 and 299 the phrase 'of roughly 10,087 million litres' and 'of about 1,285 thousand heads' are too vague in my opinion, especially when it is followed by percentages given to two decimal places, why not just state 'of 10087 million litres', 'of 1285 thousand head'. I realise there are many assumptions in this model and many inaccuracies arising from the data available but why not be more precise about what was actually found. This also occurs elsewhere in the manuscript.

Line 308 and 315 'roughly', line 316 'about 89 thousand' why not be precise?

Line 402 are these percentages an overall percentage of loss gap or percentage variation to baseline?

Line 419-424 Are you saying Bennett and IJpelaar (2005) loss estimates are higher than found here or lower? The Loss Gap estimate for the English and Welsh sectors could be higher or lower, does it matter? It is just a source of difference.

Line 444 I think the statement regarding fertility needs a reference.

6. PLOS authors have the option to publish the peer review history of their article (what does this mean?). If published, this will include your full peer review and any attached files.

Reviewer #1: No

Reviewer #2: No

Reviewer #3: No

---

## [Author Response · Author response to Decision Letter 0]

27 May 2024

Reviewer 1

Dear reviewer,

Thank you for your time and effort in reviewing our work. Your feedback is very much appreciated. 

(Moreover, the authors made scenarios which are totally unrealistic. Assuming no mortality and no vet costs is really unrealistic for dairy herds. At dairy herds, production diseases (eg mastitis, lameness, metabolic disorders) will always retain as they can't be eradicated. Comparing results to these utopian scenarios (no mortality and no vet costs) is thus extremely overestimating the losses. )

We understand the comment about the assumption to bring mortality and vet costs to zero. And yes, in the real world, removing all health hazards from dairy production systems is not possible. This theoretical approach is to assess the burden of the existing health hazards (Loss Gap) in the current British dairy production system. We are not claiming that disease will be removed. We aim to establish the boundaries to the Loss Gap, which is composed by the production performance impairment resulting from the health hazards plus the expenditure for preventing and/or controlling these health hazards. In our view, this can be achieved by comparing the current situation (that with all the health problems you have indicated, plus all the others) with a scenario where no disease is present (also removing the need for treatments and associated vet costs which are part of the burden due to disease). Having this gap established will then help us to:

• Think whether current burden estimates for specific health problems are taking into account morbidity issues (if we sum the burden from studies looking in to single-cause problems, are we going beyond the gap?)

• Understand how well we “know” the gap (what if a large proportion of the gap is left unattributed, after having “filled it in” with the burden from the know health hazards? Could a significant piece of the puzzle be missing?)

• Prioritise the major health concerns

Also, as a theoretical exercise, it can be argued that current disease management practices are limited to current resources and/or knowledge. It could be that technological and scientific development bring tools in the future that could allow for the existing diseases to be better managed or even eradicated. Understanding the total burden, in terms of production losses and health expenditure, deriving from the existing health hazards would help in addressing the points outlined above. 

(As the model is deterministic it does not allow to give insights in variation, which is a weak point. Only average values are taken into account.)

In fact, the exercise is conduct in a deterministic manner, based on average values. This was due to the data made available. Efforts were made to retrieve figures on the distribution of data without success. Having such statistical information around the figures would have added great value to this work, and this is one of the limitations identified in the paper as well as a finding. 

(small remark: replacement herd is a strange term. I think the authors mean young stock herd.)

Thank you! We have replaced the term replacement herd with young stock. 

Reviewer 2

Dear reviewer,

Thank you for your time and effort in reviewing our work. Your feedback is very much appreciated. 

(This approach would to give a global economic assessment to estimate what is the more profitable model (lower loss gap) to prevent inefficiencies regarding the different assumptions and parameters used in this work.)

Just to make sure the purpose of the work is clear we would like to clarify the output of the analysis and what it entails. By comparing the baseline (enterprise budget in the current situation) with the counterfactual scenarios (enterprise budget in the utopias), we establish the boundaries for the burden of health-related issues that are currently present in dairy production systems (by capturing the reduced production due to disease impact and the expenditure farmers have in mitigating the impacts of the disease). This approach sets the boundaries for the total health-related burden, and offers a framework that helps us to:

• Think whether current burden estimates for specific health problems are taking into account comorbidity issues (if we sum the burden from studies looking in to single-cause problems, are we going beyond the gap?)

• Understand how well we “know” the gap (what if a large proportion of the gap is left unattributed, after having “filled it in” with the burden from the know health hazards? Could a significant piece of the puzzle be missing?)

• Prioritise the major health concerns

Additionally, it offers the starting point for further economic analysis, such as cost-benefit analysis, which could indicate if a particular health intervention is logical from a financial standpoint. 

(But a more detailed justification and limitation for these 3 counterfactual scenarios can elucidate the readers. Also, a critical discussion about the limitations and constraints of this study is welcome. As example, the potential effect of farm size in each production system regarding incomes (production cost) and outcomes; or the cost of disease prevention in farms. ) 

The justification for having defined the counterfactual scenarios is provided in the discussion in lines 438-463. The limitations are also discussed here and further in lines 541-555. Data issues are identified as the main constraint to the work and are also one of the key findings. In particular, the way farms were classified (profitability and not productivity) was quite challenging. In lines 507 to 523, these issues are explored, referring to the effect of economies of scale linked to larger herds. It also highlights that by matching the milk yield based on benchmarking, the model ignores certain factors that have an influence on this production trait. 

Line 514 - We have added “investment in disease control and prevention” to the list of factors to provide clarity as suggested in the example the reviewer gave. Additionally, we have added text to clarify the reason for Utopias 2 and 3.

Lines 541-547 - Three utopian scenarios were defined as counterfactual to estimate the Loss Gap, all considering the average milk yield across all farms to be equal to the that observed in the top performing farms. Utopias 2 and 3 were developed to assess the impact of a key production factor – concentrated feed in utopia 2 – and a key market parameter – milk price in utopia 3 - in the Loss Gap. This has shed light about the impact of efficient use of key inputs and prices in the Loss Gap. Yet, the approach taken requires caution when interpreting any of the results across the different scenarios.

(Just a curiosity: if we consider the top 10% as reference what is the annual Loss Gap?)

Regarding the top 10% as reference, this would have been a good improvement to the output of this work in our perspective. This exercise is based on the assumption that animals underperform due to health and welfare issues. So, the higher the granularity in terms of farm performance the more accurate the reference. That is to say that if we could have the top performing farm for each of the production systems (classified according to calving pattern), we would compare all the other farms to that one reference, assuming that the cows kept in this farm were the ones with least health and welfare problems affecting their productivity. However, data was offered to us grouped as such – top 25%, bottom 25% and those in the interquartile range. So, we had no way to estimate the Loss Gap using other performance levels as references. 

Reviewer 3

Dear reviewer,

Thank you for your time and effort in reviewing our work. Your feedback is very much appreciated. 

(Line 354 the section describing 'Dissecting the loss gap' needs some clarification. What do the given percentages represent, variation to baseline scenario? For utopia scenario 1, addition of 65.8%, 34.2% and 11.4% is 111.4%?

In relation to this point I think it should be made clearer in the manuscript that for example a variation of 65% from base on veterinary services and medicines when they are at 1 ppl only gives a modest change in cost expressed as ppl whereas a 35% variation in herd replacement costs would be a much bigger change in ppl.)

Thank you for calling our attention to this! Indeed the percentages represent the variation from the baseline scenario. This means that cost centres can have a negative, neutral or positive variation from the baseline. In the case for Utopia 1, adding the percentages from all cost centres will return 100% as some variation will be negative. For clarity the following text has been added to the section:

Lines 364-366: The percentages presented in Fig. 4 reflect the variation of each cost centre from the baseline. This means that the variation for each cost centre from baseline to the Utopias can take a negative, a neutral (zero) or a positive value. 

Additionally, some parts were corrected as we felt that it was poorly interpreting the values in the figure. 

It is true that herd replacement costs are much considerable compared to vet services when looking at costs per litre of milk (ppl). However, the variation isn’t related to the contribution of herd replacement costs per litre of milk, but to the contribution of herd replacement costs as a whole. As already mentioned the percentages are the variation of the cost centres between baseline and utopia. So the 65% increase in vet services means that the costs with vet services in the baseline were 65% higher when compared to utopia 1. This is due to the fact that we are artificially “removing” those costs per each litre produced. Given the amount of litres produced that leads to a considerable sum. The herd replacement costs have different parameters affecting it, mortality being a key parameter. When we artificially remove mortality, we are reducing the costs in relation to the rate – 2% in case of the cows, 8% in case of the heifers. Thus, and as you have suggested, even though the mortality isn’t very high, the impact in the Loss Gap is quite significant. 

(Line 447-450 In utopia 1 the yield level achieved by the best performing herds is still constrained by disease but most likely at a much lower level. Top level performance here is not 'disease free'. I think this needs to be stated. )

Thank you! We have changed the text to make this more obvious. The text reads now as per below:

“…a dairy sector in which all animals perform as the top reference unit according empirical data, which still considers the effect of disease constraining animal performance, in a context without disease (no veterinary nor medicine costs) nor mortality. Thus, disease effects are artificially removed by bringing mortality and expenditure with vet services and medicine costs to zero.”

(The Loss Gap would be much greater if in the scenarios modelled a top performing yield level that was genuinely disease free. It is a limitation of this study that you cannot make this true disease free comparison. For example the NMR 500 (https://www.nmr.co.uk/news/nmr-500-herd-kpi-2023-report-released) herds reports a much higher yield level and could be cited as evidence. This would also suggest that the loss gap estimated in this current study is also under estimated and gives an indication of how big the under estimate is. The NMR 500 report however does not provide all the figures necessary to estimate a loss gap using the methods reported.)

Thank you. Indeed the study by Hank and Kossaibati was one of the references explored for this exercise. This issue was discussed in the data sources section, where we explain why we decided to use the AHDB as the main source of data. Part of the problem was also further explored in the discussion, where we highlight the interdependency of certain production indicators (eg. high yielding cows tend to have fertility problems). The exercise, however, is in fact limited when attempting to simulate a free-from-disease scenario. We have added text in the discussion (below) to highlight this in relation to the work conducted by hanks and kossaibati (for time scale, we have decided to keep the reference that we had explored when conducting the analysis). 

Additionally, the developed utopias are limited while attempting to simulate a production system without diseases. As explained above, the milk yield is based on empirical data, and thus the “gold standard” is still hampered by disease impact. Thus, it could be that the Loss Gap is underestimated. For example, the study by Hanks and Kossaibati (2020) indicate that cows from the best 25% farms considering yearly milk yield per cow of the 500 farms in their sample, which are also affected by health issues, produce at least 9,850 kg of milk per year {Hanks, 2020 #623}– a 12.5% difference from the average 8,749 kg milk/cow/year used in this study.

(Lines 1-4 and elsewhere e.g. line 70 uses the phrase 'yield gap' whilst the Abstract and rest of the manuscript uses 'Loss Gap'. This is slightly misleading as I think what is being evaluated here is the difference in gross margin and would suggest to use the term 'Loss Gap' throughout.)

Thank you! We have amended as suggested and where adequate. As the Loss Gap captures different costs and losses we felt that yield gap was an adequate term for some part of the text. 

(Line 58 or is the higher prevalence due to a survivor bias i.e. lower yielding cows disposed of earlier in a disease episode? And does this mean that earlier culling of those cows affects the Loss Gap. I'm guessing no information is available on cow yield at exit but interesting to speculate.)

Thank you! I would not say it is a survivor bias, at least in these systems. For example, the latest study concerning hoof health in dairy cattle in the UK indicates around 30% of lame cows at any point in time. Confinement, concrete, genetic selection, nutrition (unbalanced for the natural requirements of the cows at different stages of lactation)…all contribute to the increase of lameness levels/maintenance of high levels of lameness. Also, farmers will make decision about removing animals from their herds based on production – on average a cow in the UK is removed from the herd after 4.5 lactations. This is probably when cows start having too many problems (lameness, mastitis, fertility). And depending on the problem it could have an impact on the Loss Gap. If a farmers is replacing a high producing cow with health problems with a young cow that hasn’t reached its production peak (only after 2nd lactation) it is producing less than it could, thus increasing the Loss Gap. 

(Line 75 'of the gap')

Thank you! Dully corrected.

(Line 148 uses the abbreviation AYRC while at line 180 it is AYR.)

Thank! All have been corrected to AYRC.

(Line 294 and 299 the phrase 'of roughly 10,087 million litres' and 'of about 1,285 thousand heads' are too vague in my opinion, especially when it is followed by percentages given to two decimal places, why not just state 'of 10087 million litres', 'of 1285 thousand head'. I realise there are many assumptions in this model and many inaccuracies arising from the data available but why not be more precise about what was actually found. This also occurs elsewhere in the manuscript. Line 308 and 315 'roughly', line 316 'about 89 thousand' why not be precise?)

Thank you! We have removed these terms. 

(Line 402 are these percentages an overall percentage of loss gap or percentage variation to baseline?)

Thank you! Percentage variation from baseline. We have corrected the text to provide more clarity about what these figures are. 

(Line 419-424 Are you saying Bennett and IJpelaar (2005) loss estimates are higher than found here or lower? The Loss Gap estimate for the English and Welsh sectors could be higher or lower, does it matter? It is just a source of difference.)

The “lower boundary” of the estimates from Bennett and IJpelaar is higher than the highest estimate for the Loss Gap - £256 million vs £227 million. The purpose of the comparison is to call attention to the comorbidity issue. Could it be that by adding all of the burden from single cause diseases we are in fact overest

---

## [Decision Letter · Decision Letter 1]

17 Jun 2024

Setting the boundaries – an approach to estimate the Loss Gap in dairy cattle

PONE-D-24-02553R1

Dear Dr. Sucena Afonso,

We’re pleased to inform you that your manuscript has been judged scientifically suitable for publication and will be formally accepted for publication once it meets all outstanding technical requirements.

Kind regards,

Chisoni Mumba

Academic Editor

PLOS ONE

Additional Editor Comments (optional):

Reviewers' comments:

Reviewer's Responses to Questions

**Comments to the Author**

1. If the authors have adequately addressed your comments raised in a previous round of review and you feel that this manuscript is now acceptable for publication, you may indicate that here to bypass the “Comments to the Author” section, enter your conflict of interest statement in the “Confidential to Editor” section, and submit your "Accept" recommendation.

Reviewer #2: (No Response)

Reviewer #3: All comments have been addressed

2. Is the manuscript technically sound, and do the data support the conclusions?

Reviewer #2: Yes

Reviewer #3: Yes

3. Has the statistical analysis been performed appropriately and rigorously? 

Reviewer #2: Yes

Reviewer #3: Yes

4. Have the authors made all data underlying the findings in their manuscript fully available?

Reviewer #2: No

Reviewer #3: Yes

5. Is the manuscript presented in an intelligible fashion and written in standard English?

Reviewer #2: Yes

Reviewer #3: No

6. Review Comments to the Author

Reviewer #2: Thanks to the authors for provinding this revised version. The comments and suggestions of this reviewer were fully answered. In my opinion, the manuscript is able to be published in the present form.

Reviewer #3: Thanks to the authors for dealing with the points raised earlier. I have no further comments to make.

7. PLOS authors have the option to publish the peer review history of their article (what does this mean?). If published, this will include your full peer review and any attached files.

Reviewer #2: No

Reviewer #3: No

---

## [Editor Report · Acceptance letter]

19 Jun 2024

PONE-D-24-02553R1 

PLOS ONE

Dear Dr. Sucena Afonso, 

I'm pleased to inform you that your manuscript has been deemed suitable for publication in PLOS ONE. Congratulations! Your manuscript is now being handed over to our production team.

Kind regards, 

on behalf of

Dr Chisoni Mumba 

Academic Editor

PLOS ONE